# Antifungal Activities of L-Methionine and L-Arginine Treatment In Vitro and In Vivo against *Botrytis cinerea*

**DOI:** 10.3390/microorganisms12020360

**Published:** 2024-02-09

**Authors:** Shengwang Li, Youwei Yu, Peng Xie, Xianran Zhu, Chao Yang, Linjing Wang, Shaoying Zhang

**Affiliations:** College of Food Science, Shanxi Normal University, Taiyuan 030031, China; liswang98@163.com (S.L.); xiepeng669@outlook.com (P.X.); ych112310@163.com (C.Y.);

**Keywords:** L-methionine, L-arginine, *Botrytis cinerea*, antifungal effect, postharvest fruit and vegetables

## Abstract

Gray mold caused by *Botrytis cinerea* is a common postharvest fungal disease in fruit and vegetables. The prevention and treatment of postharvest gray mold has been one of the hot research issues addressed by researchers. This study aimed to investigate the effect of L-methionine and L-arginine on *Botrytis cinerea* in vitro and on cherry tomato fruit. The results of the in vitro experiment showed that L-methionine and L-arginine had significant inhibitory effects on the mycelial growth and spore germination of *Botrytis cinerea*, and the inhibitory effects were enhanced with increasing L-methionine or L-arginine concentration. In addition, L-methionine and L-arginine treatment increased the leakage of *Botrytis cinerea* electrolytes, proteins and nucleic acids. The experiment involving propidium iodide staining and malondialdehyde content assay also confirmed that L-methionine and L-arginine treatment could lead to cell membrane rupture and lipid peroxidation. The results of scanning electron microscopy further verified that the morphology of hyphae was damaged, deformed, dented and wrinkled after treatment with L-methionine or L-arginine. Fruit inoculation experiments displayed that L-methionine and L-arginine treatments significantly inhibited the occurrence and development of gray mold in postharvest cherry tomato. Therefore, treatment with L-methionine or L-arginine might be an effective means to control postharvest gray mold in fruit and vegetables.

## 1. Introduction

*Botrytis cinerea* is a highly prevalent detrimental fungus in a wide range of fruit and vegetables. It has the characteristics of easy adaptability, fast growth and spore production [1]. More than 1000 plant species have been reported to be infected by *B. cinerea*, especially fruit and vegetables with a high water content, such as grapes, tomatoes, strawberries and peppers [2]. Gray mold can lead to serious economic losses to global fruit and vegetable industry during harvesting, storage, processing and transportation. At present, gray mold ranks among the top 10 most significant fungal plant diseases [3]. Nowadays, the control of postharvest disease mainly depends on the application of chemical fungicides [4].

Conventional synthetic chemical fungicides such as iprodione [5,6], anilinopyrimidines [7,8] and pyrimethanil [9,10] have been used to control gray mold by disrupting the formation of fungal cell membranes. However, the extensive use of synthetic fungicides adversely impacts the ecosystem and human health, owing to the accumulation and persistence of residues. In Europe, dicarboximide fungicides, including iprodione, glyphosate and procymidone, are no longer approved [11,12]. Nevertheless, the fungicide resistance issue is becoming increasingly problematic as populations resistant to mainstream selective fungicides have emerged in the field, limiting the widespread application of fungicides [13,14]. Thus, there is an urgent need to explore environmentally friendly substances that can effectively control gray mold. In recent years, a large number of substances considered safe have been successfully used to control postharvest gray mold damage in fruit and vegetables, such as CO [15,16], propionate [17,18] and sodium silicate [19,20]. Additionally, phenolic acids derived from various fruit and vegetables, such as ferulic acid [21,22] and chlorogenic acid [23,24], along with specific essential oils like Origanum Dictamnus [25,26] and thyme [27,28] were employed to mitigate the damage caused by gray mold disease. Amino acids, inherently present in plants as natural products, play essential roles in various intracellular metabolic pathways, including ATP generation, nucleotide synthesis and redox homeostasis [29]. The important role of amino acids is not limited to this, as many previous studies have revealed that they also have the ability to induce disease resistance in plants [30]. For instance, L-glutamic enhanced the resistance of pear fruit to *Penicillium expansum* and induced blast resistance in rice [31,32]. Phenylalanine treatment boosted mango resistance to both cold injury and anthracnose during storage [33,34]. Additionally, L-cysteine was beneficial to preserve fresh-cut red beets by increasing the contents of gallic acid, caffeic acid, chlorogenic acid, kaempferol and betalains [35]. Furthermore, in one study cysteine inhibited the activities of polyphenol oxidase enzymes (PPO) and peroxidase enzymes (POD), enhancing the quality of lychee under refrigerated conditions [36].

L-methionine (Met), an indispensable amino acid, plays a crucial role in protein synthesis. It is also beneficial to maintaining oxidative balance, owing to its antioxidant capacity. L-methionine and its metabolite s-adenosylmethionine (SAM) serve as methyl donors, which is a function necessary for normal cell metabolism [37]. Their metabolites are ethylene, polyamines (PAs), spermidine and spermine. Ethylene may stimulate fruit and vegetable senescence [38], while spermidine and spermine have anti-aging effects [39]. L-methionine enhances or inhibits aging, depending on its transformation kinetics. L-arginine (Arg), a structural unit of protein, participates in a variety of metabolic pathways and produces many signaling molecules. It can be converted into polyamines (PAs), γ-aminobutyric acid (GABA) and proline (Pro), improves the activity of nitric oxide synthase (NOS) and promotes the production of nitric oxide (NO). L-arginine has strong immunomodulatory effects and inhibition ability against pathogens. Its immunomodulatory effects have been widely reported [40]. Both L-methionine and L-arginine have strong antioxidant properties. L-arginine has been reported to inhibit the tissue browning of apples, lettuce [41] and broccoli [42], and to enhance antioxidant activity and alleviate postharvest rot in pomegranates [43]. This has been demonstrated in published studies of whole produce. In addition, L-methionine might protect the stability of anthocyanins in bayberry [44] and inhibit the browning of lychee tissue [45]. In general, the research of these two amino acids (L-methionine and L-arginine) has mainly focused on improving the postharvest resistance of fruit and vegetables, but their antifungal activities and inhibitory mechanisms in vitro have not been reported.

In this study, L-methionine and L-arginine were used to control gray mold, and the potential mechanism was also investigated. This study included an evaluation of the effects of L-methionine and L-arginine treatments on *B. cinerea* colony growth, spore germination, cell morphology, cell membrane permeability and the inhibition of the development of gray mold in cherry tomatoes. This research could lay the ground for understanding the antifungal activity of L-methionine and L-arginine and provide potentially sustainable alternatives for the prevention and management of gray mold disease of fruit and vegetables postharvest.

## 2. Materials and Methods

### 2.1. Pathogen and Spore Suspension

#### Source of Fungi and Preparation of Spore Suspension

The strain of *Botrytis cinerea* (No: BNCC185786) was purchased from BeNa Biological Co., Ltd. (Beijing, China). *Botrytis cinerea* was cultured at 28 °C for two weeks on potato dextrose agar (PDA) plate medium. Then, the PDA plate was flooded with sterile distilled water to collect spores. Subsequently, the spore suspension was filtered through four layers of sterile gauze to remove some mycelium fragments. The concentration of the spore suspension was adjusted according to experimental requirements, and the suspension was used immediately after preparation [2]. Its concentration was determined using a hemocytometer method with an optical microscope (Olympus CX31, Olympus Co., Ltd., Tokyo, Japan).

### 2.2. Chemicals and Fruit

The biochemical reagents L-methionine and L-arginine (both ≥99%) were obtained from Aladdin Biotechnology Co., Ltd. (Shanghai, China). Cherry tomatoes (*Solanum lycopersicum* var. *Saopola*) were harvested at commercial maturity and purchased from the Huilong Farmers’ Market located in Yuci District, Jinzhong City, Shanxi Province, China.

### 2.3. Effect of L-Methionine and L-Arginine on the Mycelial Growth of B. cinerea

The effect of L-methionine and L-arginine on the mycelia growth of *B. cinerea* in vitro was assayed using the method described by Yang et al. [1]. In detail, L-methionine and L-arginine were added to Petri dishes (90 mm diameter) containing potato dextrose agar, yielding L-methionine media with a concentration gradient of 0, 0.5, 5, 50 and 100 mmol L^−1^ and L-arginine media with a concentration gradient of 0, 6.25, 12.5, 25, 50 and 100 mmol L^−1^. After the PDA solidified, a sterile punch was used to create the mycelial disks (13 mm) from a pathogen culture. The mycelial disks were placed in the center of each Petri dish and subsequently incubated at 25 °C. The colony diameter of *B. cinerea* was measured daily. The investigation was repeated three times.

The effects of L-methionine and L-arginine on the spore germination of *B. cinerea* in vitro were assessed according to previously reported methods [46]. L-methionine solutions at concentrations of 0, 0.5, 5, 50 and 100 mmol L^−1^ and L-arginine solutions at concentrations of 0, 6.25, 12.5, 25 and 50 mmol L^−1^ were prepared in 20 mL of sterile potato dextrose broth medium (PDB).

Subsequently, 50 μL of *B. cinerea* spore suspension (1 × 10^6^ spores mL^−1^) was inoculated into the PDB. The culture was incubated on a rotary shaker at 25 °C for 12 h, washed with phosphate-buffered saline (PBS, pH 7.0) at least twice, and then resuspended in sterile water. Subsequently, a volume of 20 μL from the spore suspension was carefully applied onto a concave slide using a sterile pipette. The germination rate and germ tube elongation of *B. cinerea* were then assessed by counting approximately 200 spores under a light microscope.

### 2.4. Assay of L-Methionine and L-Arginine on Extracellular Electrical Conductivity, Protein Leakage and MDA Content

#### 2.4.1. Determination of Extracellular Electrical Conductivity

The effects of L-methionine and L-arginine on the extracellular conductivity of *B. cinerea* were assayed using previously reported methods, with minor modifications [47]. A 100 µL amount of *B. cinerea* spore suspension (1 × 10^6^ mL^−1^) was cultivated in 100 mL of PDB medium on a rotary shaker (130 rpm/min) at 25 °C for 48 h. The culture was centrifuged at 4000× *g* at 4 °C for 10 min. The mycelium was washed at least twice with 50 mmol L^−1^ PBS (pH 7.0). Afterward, mycelia (3 g) were added to 40 mL of sterile water containing 0, 50 or 100 mmol L^−1^ L-methionine solution or 0, 25 or 50 mmol L^−1^ L-arginine solution. Mycelia were filtered at 0, 2, 4, 6 and 8 h using a double layer of clean cheesecloth. Subsequently, a conductivity meter was utilized to measure the permeability and extracellular conductivity of the filtered suspension (DDS-307; Shanghai Leici Instrument Co., Ltd., Shanghai, China). Each treatment was replicated independently three times.

#### 2.4.2. Measurement of Nucleic Acids and Proteins

The release of nucleic acid and protein from *B. cinerea* was conducted according to the method described by Kong et al. [48]. Aliquots of spore suspension (100 µL, 1 × 10^6^ spores mL^−1^) were added to PDB containing varying concentrations of L-methionine and L-arginine. After incubating at 28 °C for 2, 4, 6 or 8 h, the treated spores were centrifuged at 5000× *g* for 10 min. The supernatants were analyzed. The release of nucleic acids and proteins from the cytoplasm was measured using a microplate reader (The Spectra Max M2, Molecular Devices, Inc., Sunnyvale, CA, USA) by detecting the absorbance values at 260 nm and 280 nm, respectively. To eliminate potential background absorbance effects from L-methionine and L-arginine, the absorbance values of blanks were subtracted from the final results. The experiment was repeated three times.

#### 2.4.3. Assay of Malondialdehyde (MDA)

The lipid peroxidation of *B. cinerea* plasma membranes was assayed by determining the malondialdehyde (MDA) content using the thiobarbituric acid method [49]. The spore suspension (100 μL) with a concentration of 1 × 10^6^ mL^−1^ was inoculated into 50 mL of PDB medium on a rotary incubator (SHA-B, Tianjin Saidelisi Instrument Inc., Tianjin, China) at 130 rpm/min for 3 d at 28 °C. The mycelia were harvested in a paper filter and washed twice with sterile distilled water. Subsequently, the mycelia were resuspended in 50 mL of 0, 50 or 100 mmol L^−1^ L-methionine solution and 0, 50 or 100 mmol L^−1^ L-arginine solution. The MDA content of mycelia was determined after incubation on a rotary incubator at 28 °C for 0, 4, 8, 12 and 16 h. The experiments were repeated thrice.

### 2.5. Measurement of Plasma Membrane Integrity

The spore suspension (100 μL) was transferred to 250 mL conical flasks containing 100 mL of PDB and incubated at 130 rpm and 28 °C. Mycelia were collected after incubation for 3 d and thoroughly washed thrice with sterile water to remove residual medium. The mycelia (0.2 g) were resuspended in 50 mL of 0 (control), 50 or 100 mmol L^−1^ L-methionine and 50 or 100 mmol L^−1^ L-arginine solution and incubated at 28 °C with constant shaking (130 rpm) for 12 h. Afterward, the mycelium culture was centrifuged (5000× *g*, 10 min) at 4 °C, the supernatant was poured out, and the mycelium was rinsed 3 times with PBS. It was centrifuged after each rinse. Afterwards, propidium iodide (PI) was added for staining observation.

### 2.6. Scanning Electron Microscopy (SEM)

Scanning electron microscopy was carried out to investigate the effects of L-methionine and L-arginine on the morphological changes of *B. cinerea* hyphae, according to previously reported methods [50]. In brief, the mycelial disks were placed in 100 mL of PDB medium containing sterile distilled water (as control); 5, 50 or 100 mmol L^−1^ L-methionine solution; and 6.25, 12.5 or 25 mmol L^−1^ L-arginine solution. Subsequently, the samples were incubated in shake flasks at 25 °C for 48 h. Mycelia were collected, rinsed at least twice with PBS (pH 7.0) and then resuspended in PBS buffer. The mycelia were fixed in 2.5% glutaraldehyde for 4 h and washed three times with the same phosphate buffer. Then, the samples were transferred to a series of ethanol solutions (30%, 50%, 70%, 85%, 95% and 100%) for 20 min each, followed by 100% ethanol for 20 min. After drying in a freeze-drying apparatus, the samples were sputter-coated with gold and observed with a scanning electron microscope (JSM-IT800, JEOL, Ltd., Tokyo, Japan).

### 2.7. Antifungal Effect of L-Methionine and L-Arginine on Cherry Tomato Inoculated with B. cinerea

The effect of L-methionine and L-arginine treatment on *B. cinerea* in vivo was assessed using a previously reported method [51]. After they were disinfected on the surface with 0.01% sodium hypochlorite for 2 min, cherry tomato fruit were air-dried. Subsequently, wounds (3 mm deep and 3 mm wide) were created at the equator of each fruit using a sterilized stainless steel nail. Finally, 5 μL of *B. cinerea* spore suspension (1 × 10^6^ spores mL^−1^) was injected into each wound. The fruit was air-dried for 1 h, and then 20 μL of L-methionine at 0.5, 5, 50 or 100 mmol L^−1^ or L-arginine at 6.25, 12.5, 25 or 50 mmol L^−1^ was added into each wound. Sterile distilled water served as a control sample. Following air-drying for 1 h at room temperature, the treated fruit were transferred into plastic containers with sterile water to sustain a humidity level of 95%. Subsequently, they were stored at 22 °C. The lesion diameter beyond each wound was measured 4 d after inoculation. Each treatment contained three replicates with 30 fruit per replicate.

### 2.8. Statistical Analysis

The data were subjected to analysis using SPSS version 25 statistical software (SPSS, Chicago, IL, USA) in order to perform an analysis of variance (ANOVA). All the experiments were performed in triplicate (*n* = 3). To compare the different treatments, Duncan’s multiple range test was employed at a confidence level of 95% (*p* < 0.05)

## 3. Results

### 3.1. Effects of L-Methionine and L-Arginine on the Mycelial Growth and Spore Germination of B. cinerea

The effects of L-methionine and L-arginine on the mycelial growth of *B. cinerea* are shown in Figure 1. After *B. cinerea* was cultured for 4 d at 25 °C, the colony of the control group almost expanded to the entire PDA plate, but both L-methionine and L-arginine inhibited the mycelial growth of *B. cinerea*. When the concentration of L-methionine or L-arginine increased from 0 to 100 mmol L^−1^, the inhibitory effects were gradually enhanced. At 4 d, the colony diameter of the control group was 8.93 cm. Compared with the control sample, the colony diameters of the treatment groups were reduced by 17.1%, 19.3%, 34% and 36.1% at the L-methionine concentrations of 0.5, 5, 50 and 100 mmol L^−1^, respectively. The colony diameters of the treatment groups were reduced by 10%, 28.6%, 51.9%, 72.6% and 100% at the L-arginine concentrations of 6.25, 12.5, 25, 50 and 100 mmol L^−1^, respectively. At the L-arginine concentration of 100 mmol L^−1^, the mycelial growth of treated *B. cinerea* was completely inhibited (*p* < 0.05). These results indicate that both L-methionine and L-arginine treatments could inhibit the mycelial growth of *B. cinerea*. At the same treatment concentration, the antifungal effect of L-arginine was higher than that of L-methionine.

Fungal spores have remarkable tolerance and adaptability to various stressful situations, enabling their survival in highly challenging ecological conditions. The effective inhibition of fungal spore germination and germ tube elongation by antimicrobial agents is of great significance for the prevention and treatment of postharvest diseases. As shown in Figure 2, after treatment with L-methionine and L-arginine, both spore germination and germ tube elongation of *B. cinerea* in PDB medium were inhibited. After 15 h, almost all the spores of the control group were germinated, but L-methionine and L-arginine treatment inhibited the spore germination and germ tube elongation of *B. cinerea*. Higher concentrations showed a better inhibition effect. As for the spore germination rate of *B. cinerea*, that of the control sample was 97.3%, that of the samples treated with 100 mmol L^−1^ L-methionine was 17.3% lower than that of the control sample, and that of the samples treated with 50 mmol L^−1^ L-arginine was 78.3% lower than that of the control sample. These results indicate that L-arginine had certain dose-dependent inhibitory effects on the germination of *B cinerea* spores. At the same concentration, the inhibitory effect of L-arginine on spore germination was higher than that of L-methionine.

### 3.2. Effects of L-Methionine and L-Arginine on Extracellular Conductivity, the Leakage of Nucleic Acid and Protein and the MDA Content of B. cinerea

As shown in Figure 3, the electrical conductivity of the control group changed little during the 8 h culture period, increasing only to 22%. After treatment with 50 and 100 mmol L^−1^ L-methionine for 8 h, the electrical conductivity changes were 75.1% and 85.6%, representing increases of 53.1% and 63.6% compared with the control samples, respectively. After 8 h treatment with 25 mmol L^−1^ and 50 mmol L^−1^ L-arginine, the change rates of the electrical conductivity were 87.4% and 105.4%, representing respective increases of 65.4% and 83.4% compared with the control samples. The change rate of the 50 mmol L^−1^ L-arginine treatment group was higher than that of the other treatment groups. The results show that L-methionine and L-arginine treatments increased the conductivity of *B. cinerea*.

As illustrated in Figure 4, the absorbance at 260 nm and 280 nm of *B. cinerea* treated with L-methionine or L-arginine increased with increasing processing time. This indicated that the release of *B. cinerea* nucleic acids and proteins also increased. After 8 h treatment with L-methionine and L-arginine, OD_260 nm_ and OD_280 nm_ increased in a dose-dependent manner. The nucleic acid leakage of *B. cinerea* treated with L-methionine or L-arginine showed a steadily increasing trend, while the control group had little change. The OD_260 nm_ values of samples treated with 25 and 50 mmol L^−1^ L-arginine were 1.43- and 1.56-fold higher than those of control samples, respectively. The OD_260 nm_ values of samples treated with 50 and 100 mmol L^−1^ L-methionine were 1.26 and 1.45 times higher than that of the control group. The OD_280 nm_ values of samples treated with L-arginine were 1.34- and 1.53-fold higher than those of the control group. And the values of samples treated with L-methionine were 1.31 and 1.47 times higher than that of the control group. The results of this study confirm that the treatment with L-methionine and L-arginine induced significant impairment to the cellular membrane of *B. cinerea*. Consequently, this impairment led to a substantial release of electrolytes, nucleic acids and proteins contained within the cytoplasm.

MDA is the main product of membrane lipid peroxidation, and a high content of MDA indicates a serious degree of cell membrane damage. Figure 5 exhibits the effects of L-methionine and L-arginine treatments on MDA content. Compared with the control group, the MDA contents of samples treated with the two amino acids were significantly higher. After treatment with 50 or 100 mmol L^−1^ L-arginine for 16 h, the MDA content increased sharply, to levels 4 and 3.1 times greater than those of the control group, respectively. After 16 h, the MDA contents of samples treated with 50 and 100 mmol L^−1^ L-methionine were 2.9 and 3.1 times greater than those of the control sample. Therefore, L-methionine and L-arginine treatment might cause membrane peroxidation, inhibiting the normal growth of *B. cinerea* to a certain extent.

### 3.3. Effects of L-Methionine and L-Arginine on Membrane Integrity

The cell membrane plays an important role in maintaining the balance of the fungal internal environment. Propyl iodide can cross damaged cell membranes and enter cells, specifically binding to DNA and fluorescing red. Extensive cell death could be observed in the hyphae of *B. cinerea* using the PI staining marker. As illustrated in Figure 6, after 24 h, the mycelia of the control group showed weak red fluorescence. However, the samples treated with 50 mmol L^−1^ L-methionine or 25 mmol L^−1^ L-arginine showed small areas of bright red fluorescence, while those treated with 100 mmol L^−1^ L-methionine or 50 mmol L^−1^ L-arginine displayed large areas of intense red fluorescence. There was a dose effect, such that the higher the amino acid concentration, the more mycelial death. These results show that L-methionine and L-arginine could damage the cell membrane integrity of *B. cinerea*. Moreover, the fluorescence intensity of the L-methionine treatment group was slightly lower than that of the L-arginine treatment group, indicating that L-arginine induced greater damage to the mycelia cell membrane.

### 3.4. Effects of L-Methionine and L-Arginine Treatments on Mycelial Morphology of B. cinerea

Figure 7 shows the effects of L-methionine and L-arginine treatments on the mycelial morphology of *B. cinerea*. After incubation at 25 °C, the hyphae of control samples showed a typical linear shape, regular density and smooth texture, whereas the mycelial morphology of samples treated with L-methionine and L-arginine was different from that of the control group. The hyphae of samples treated with 5 mmol L^−1^ L-methionine showed slight shrinkage, folding and deformation; the mycelial folding and deformation of samples treated with 100 mmol L^−1^ L-methionine were more significant. It was observed that the higher the concentration of the two amino acids, the more serious the deformation of the mycelia. Further, the mycelia of samples treated with L-arginine showed more significant contraction, distortion, depression and deformation than the mycelia of samples treated with L-methionine.

### 3.5. Effects of L-Methionine and L-Arginine Treatments on the Lesion Expansion of B. cinerea in Cherry Tomato Fruit

After inoculation with *B. cinerea*, cherry tomatoes were held at 25 °C for 4 d. The lesion size of gray mold significantly decreased with increasing L-methionine and L-arginine concentration (Figure 8). Compared with the control group, the lesion diameter and disease incidence of cherry tomatoes treated with 100 mmol L^−1^ L-methionine were reduced by 75.3% and 51.13%, respectively. The lesion diameter and disease incidence of cherry tomatoes treated with 50 mmol L^−1^ L-arginine were reduced by 74.2% and 29.68%, respectively. Therefore, L-methionine and L-arginine treatments significantly inhibited the onset of gray mold and the spread of lesions in postharvest cherry tomato (*p* < 0.05).

## 4. Discussion

Amino acids, the basic components of protein, are important nutrients in plants, playing an important role in various physiological processes, such as plant growth, development and dynamic homeostasis. They play a crucial part in signal transduction and defense as well [52]. In recent years, increasing attention has been paid to the application of different amino acids in improving plant disease resistance, inhibiting the development of diseases and the biosynthesis of mycotoxins [53]. L-arginine can inhibit the biosynthesis of various mycotoxins, improve disease resistance and maintain the quality of postharvest fruit and vegetables. Previous studies have shown that L-arginine treatment can improve NOS activity and the endogenous NO content of tomato fruit and activate the defense response of tomato fruit [54]. It can also inhibit the hardness decline and anthocyanin synthesis of strawberries, thus prolonging the storage period of strawberries [55]. In addition, it has also been reported that methionine can induce the expression of defense genes in fruit and vegetables, thereby inhibiting the infection of pathogens and the development of plant diseases [56,57]. In our experiment, we found that L-methionine and L-arginine treatments could inhibit the mycelial growth and spore germination of *B. cinerea*. The inhibitory effect of L-arginine was significantly better than that of L-methionine. Moreover, L-methionine had no significant inhibitory effect on the germ tube elongation of *B. cinerea* (Figure 2). As shown in Figure 1 and Figure 2, the inhibitory effects on the mycelial growth and spore germination of *B. cinerea* were dose-dependent. The findings conflict with earlier research suggesting that the two amino acids could control the decay of fruit and vegetables through their immunomodulatory effects [54,56,57,58]. It has also been found that L-arginine-rich antimicrobial peptides and surfactants can easily penetrate cell membranes or change membrane function. The possible reason is that these substances (e.g., L-arginine or lysine residues) have positively charged residues that can bind to negatively charged cell membranes and interfere with their function [59]. Based on the above reasons, the antifungal effect of L-arginine may also be owing to its positive charge, which destroys the structure and function of the cell membrane in *B. cinerea*.

To further reveal the effects of L-methionine and L-arginine on the cell membrane of *B. cinerea*, the changes in mycelial conductivity, nucleic acid and protein dissolution and mycelial morphology were measured. Compared with the control group, the mycelial conductivity change rate and OD_260 nm_ and OD_280 nm_ values were significantly increased in the treatment groups. This indicates that L-methionine and L-arginine treatments increased the membrane conductivity and nucleic acid and protein leakage of *B. cinerea* (Figure 3 and Figure 4). The results of PI staining also show that the mycelia treated with L-methionine and L-arginine had stronger red fluorescence than the control sample. Moreover, the effects were dose-dependent. This indicates that the membrane integrity of hyphae in *B. cinerea* was damaged. The reason for the destruction of the cell membrane might have been lipid peroxidation triggered by L-methionine or L-arginine treatment, which eventually led to the collapse of the cell membrane. MDA is one of the main products of lipid peroxidation in cell membranes [60]. After treatment with L-methionine or L-arginine, there was a large amount of MDA accumulation in *B. cinerea.* This verifies that treatment with L-methionine or L-arginine can destroy the integrity of cell membranes. These results are consistent with the previous experimental results. After treatment with L-methionine or L-arginine, the results of scanning electron microscopy also confirmed that the hyphae of *B. cinerea* showed irregular deformation, roughness, depression and dry surface. To realize the commercial application of L-methionine and L-arginine, it is necessary to further study the antifungal activity of L-methionine and L-arginine in practical applications. In this study, we found that L-methionine and L-arginine treatments significantly inhibited the occurrence and development of gray mold in cherry tomatoes. The result was similar to that of Zhang et al. [58]. Moreover, the control effect on gray mold in cherry tomatoes was enhanced with increasing L-methionine or L-arginine concentration. Additionally, at the same concentration (50 mmol L^−1^) of L-methionine or L-arginine, the disease diameter and incidence of fruit treated with L-methionine were lower than those of fruit treated with L-arginine. This result is different from those obtained in the in vitro experiment. This might be owing to the fact that the characteristics of plant materials determine their sensitivity to the two amino acids under study. Additionally, the abilities of L-methionine and L-arginine to trigger defense responses in plant tissues are probably different. Therefore, the application of L-methionine or L-arginine treatment for the storage and preservation of fruit and vegetables should be selected according to the specific situation.

## 5. Conclusions

L-methionine and L-arginine showed strong inhibitory effects on the mycelial growth and spore germination of *B. cinerea* in a concentration-dependent manner. Treatment with L-methionine or L-arginine promoted the lipid peroxidation of cell membranes and increased the leakage of electrolytes, proteins and nucleic acids from *B. cinerea* cells. The hyphae of *B. cinerea* treated with L-methionine or L-arginine were distorted, dented and crumpled, as observed through SEM observation. In summary, the inhibitory effects of L-methionine and L-arginine on *B. cinerea* were caused by destroying the structure of cell membranes, resulting in the leakage of intracellular substances and inhibiting germination and mycelial growth. Moreover, L-methionine and L-arginine treatments significantly inhibited the occurrence and development of gray mold in postharvest cherry tomato. Therefore, L-methionine and L-arginine are expected to be promising tools to control the postharvest gray mold of fruit and vegetables.

## Figures and Tables

**Figure 1 microorganisms-12-00360-f001:**
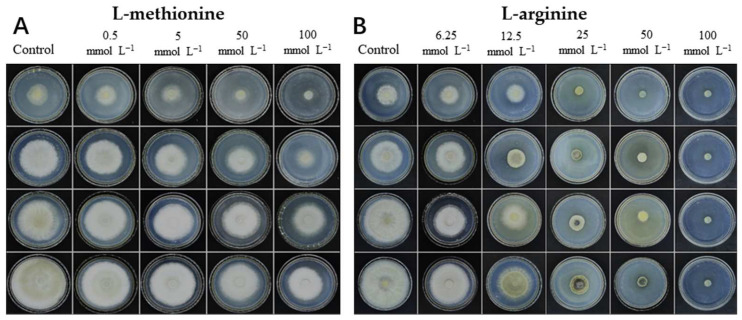
Effects of L-methionine and L-arginine treatments on mycelial growth of *B. cinerea*. In vitro growth comparison of *Botrytis cinerea* at different concentrations of (**A**) L-methionine and (**B**) L-arginine on PDA medium. Statistical analysis of colony diameters of *Botrytis cinerea* treated with (**C**) L-methionine and (**D**) L-arginine. Data are means of three experiments and vertical bars represent standard deviations. Different lowercase letters indicate significant differences among treatments evaluated at the same time, according to Duncan’s test (*p* < 0.05).

**Figure 2 microorganisms-12-00360-f002:**
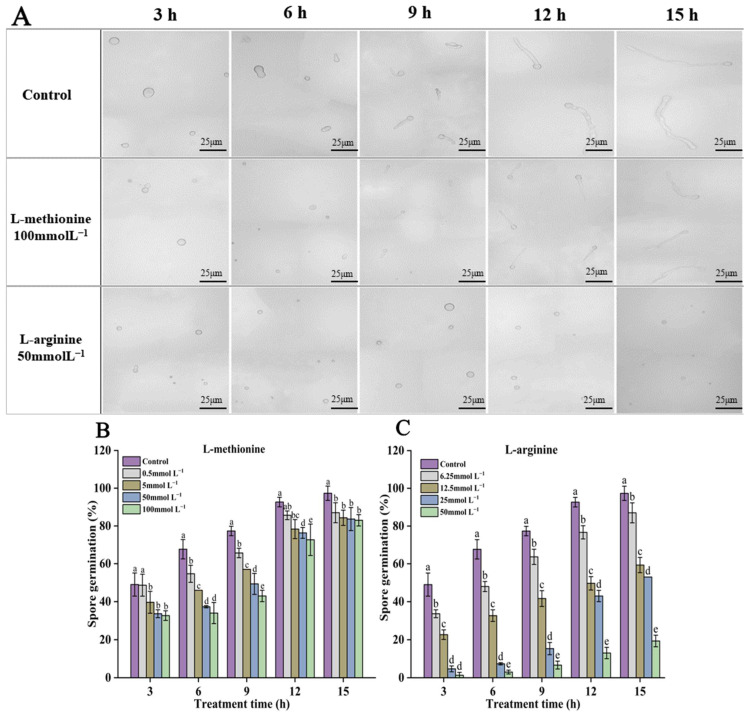
Effects of L-methionine and L-arginine treatments on spore germination of *B. cinerea*. (**A**) Representative photographs of the germinating spores of *B. cinerea* cultured with L-methionine and L-arginine at 3 h, 6 h, 9 h, 12 h and 15 h. (**B**,**C**) Statistical analysis of spore germination rate of *B. cinerea* respectively treated with L-methionine and L-arginine. Data are means of three experiments and vertical bars represent standard deviations. Different lowercase letters indicate significant differences among treatments within the same time, according to Duncan’s test (*p* < 0.05).

**Figure 3 microorganisms-12-00360-f003:**
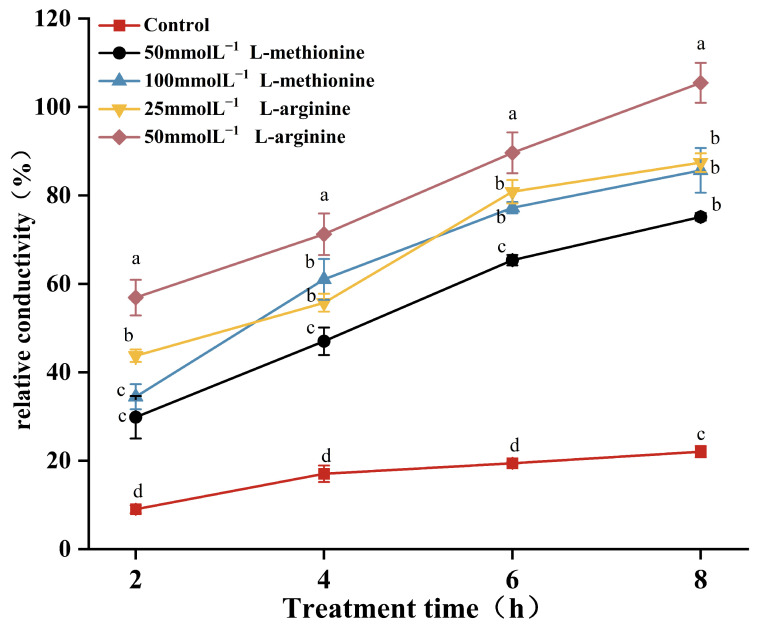
Effects of L-methionine and L-arginine treatments on extracellular conductivity of *B. cinerea*. Data are means of three experiments and vertical bars represent standard deviations. Different lowercase letters indicate significant differences among treatments within the same time, according to Duncan’s test (*p* < 0.05).

**Figure 4 microorganisms-12-00360-f004:**
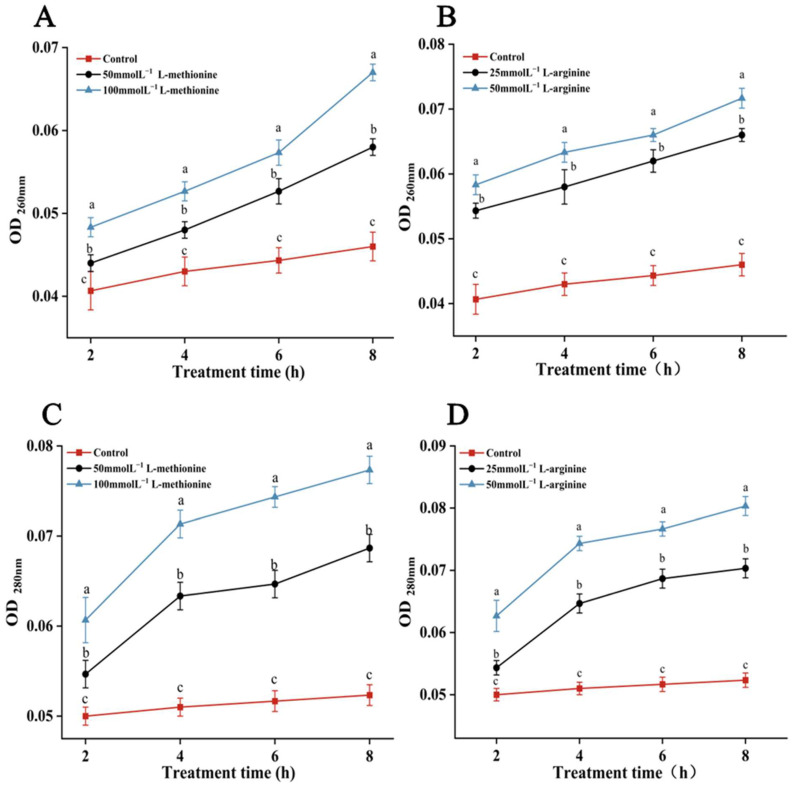
Effects of L-methionine and L-arginine treatments on nucleic acid and protein leakage of *B. cinerea*. The leakage of nucleic acids of *B. cinerea* treated with (**A**) L-methionine and (**B**) L-arginine. The leakage of proteins of *B. cinerea* treated with (**C**) L-methionine and (**D**) L-arginine. Data are means of three experiments and vertical bars represent standard deviations. Different lowercase letters indicate significant differences among treatments within the same time, according to Duncan’s test (*p* < 0.05).

**Figure 5 microorganisms-12-00360-f005:**
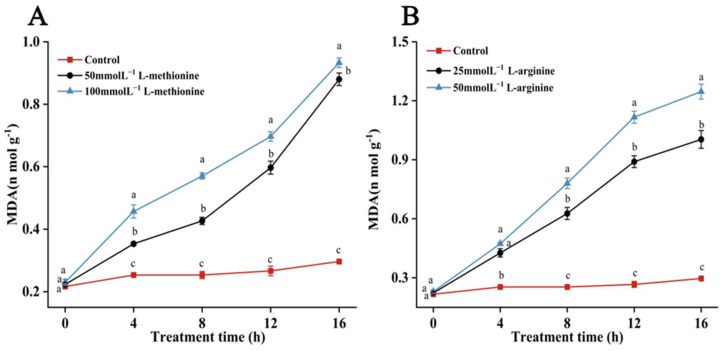
Effects of (**A**) L-methionine and (**B**) L-arginine treatments on MDA content in *B. cinerea*. Data are means of three experiments and vertical bars represent standard deviations. Different lowercase letters indicate significant differences among treatments within the same time, according to Duncan’s test (*p* < 0.05).

**Figure 6 microorganisms-12-00360-f006:**
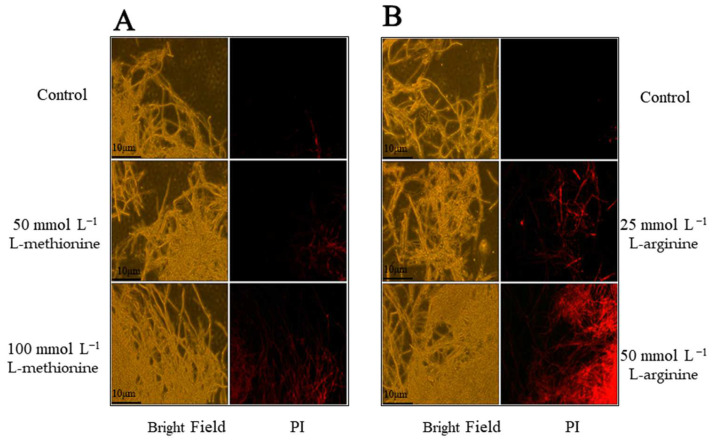
Effects of (**A**) L-methionine and (**B**) L-arginine on membrane integrity.

**Figure 7 microorganisms-12-00360-f007:**
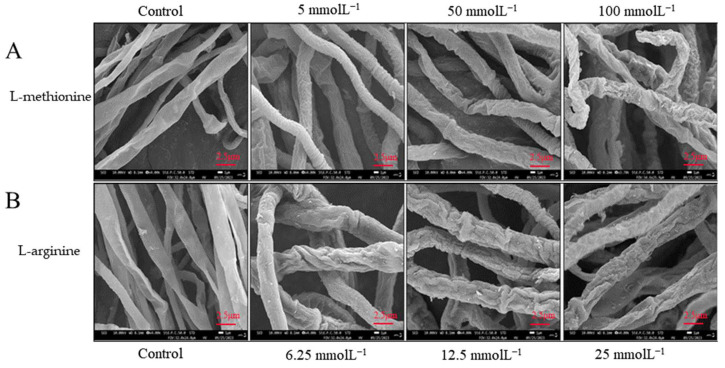
Effect of (**A**) L-methionine and (**B**) L-arginine treatments on mycelial morphology of *B. cinerea*.

**Figure 8 microorganisms-12-00360-f008:**
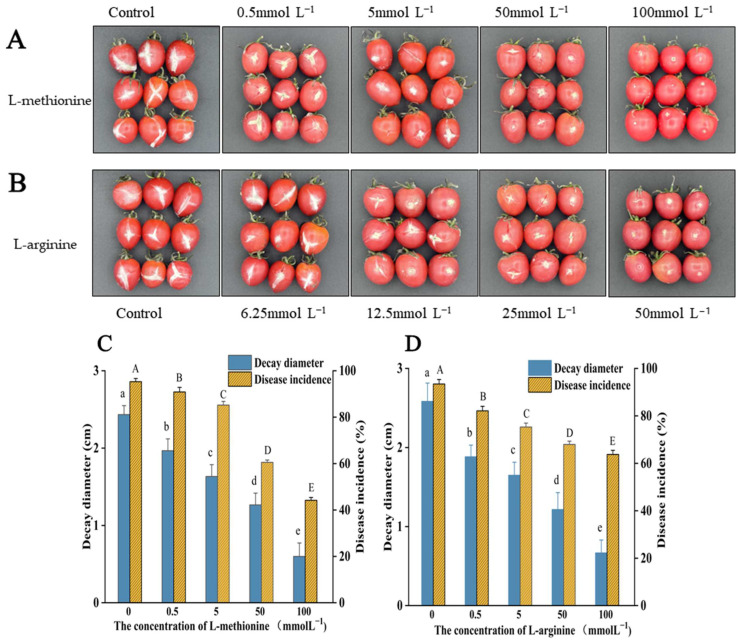
Effects of L-methionine and L-arginine treatments on the lesion expansion of *B. cinerea* in postharvest cherry tomatoes. The disease spots of cherry tomatoes treated with (**A**) L-methionine and (**B**) L-arginine. Decay diameter and disease incidence in cherry tomatoes treated with (**C**) L-methionine and (**D**) L-arginine. Data are means of three experiments and vertical bars represent standard deviations. Different lowercase letters indicate significant differences among treatments with L-methionine within the same time, according to Duncan’s test (*p* < 0.05). Different capital letters indicate significant differences among treatments with L-arginine within the same time, according to Duncan’s test (*p* < 0.05).

## Data Availability

Data are contained within the article.

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
