# Peer review of "Antifungal Activities of L-Methionine and L-Arginine Treatment In Vitro and In Vivo against Botrytis cinerea"

_microorganisms, 2024, doi:10.3390/microorganisms12020360_

Round 1

Reviewer 1 Report

Comments and Suggestions for Authors

Materials and methods. there are many unclear points: replicates/repetitions of the experiments.

In my opinion, the statistical analysis section is too confusing and not clear and some figures lack of statistical analysis.  For this, I cannot review the results paragraph.

The captions/legends of tables and figures must be rewritten more clearly, and they must be self-explanatory and written in comprehensible and correct English.

Below, the authors can find in detail some suggestions for improving the manuscript and other critical issues I have found.

Abstract.

‘This study aimed to investigate the inhibitory effect of L-methionine or L-arginine on Botrytis cinerea.’ With’ This study aimed to investigate the effect of L-methionine or L-arginine on Botrytis cinerea in vitro and on cherry tomato fruit’.’

‘had obvious inhibitory effects’, please, avoid expressions like ‘obvious’. Not suitable for a scientific paper.

Introduction

Change ‘fungus on a wide range of fruit and’ with ‘fungus on a wide range of fruits and’

Change ‘plants with a high water content’ with ‘plants with a high-water content’

Do the authors refer to ‘plant’ or perishables fruit/vegetables with high-water content’?

 ‘It has the characteristics of strong adaptability, fast growth and spore production [1] ‘

Adaptability to what? I suggest ‘Easy adaptability’

I suggest to change ‘Gray mold, caused by the pathogenic fungus B. cinerea, leads to serious economic losses to global fruit and vegetable industry during harvesting, storage, processing, and transportation.’  with ‘Gray mold can lead to serious economic losses to global fruit and vegetable industry during harvesting, storage, processing, and transportation.’

Not truly correct ‘At present, B. cinerea ranks among the top 10 most significant fungal plant diseases [3].’ B. cinerea is the microorganism, gray mold is the disease. Please, rephrase.

‘On one hand, it has ubiquitous harmfulness to fruit and vegetables;’ a concept already reported above.

‘…on the other hand, it is a model for in-depth analysis of the molecular mechanism of necrotrophic fungi pathogenicity and the development of environmentally friendly disease control strategies.

Avoid such narration or rephrase. ‘Thus, many researchers pay attention to the study of B. cinerea’ Please, check also throughout the text.

‘Owing to cost-effectiveness and ease of application, conventional synthetic chemical fungicides such as dicarboximides [5] and anilinopyrimidines [6] are commonly utilized to mitigate post-harvest decay in fruit and vegetables.’ Where? For example, in Europe, dicarboximide fungicides such as iprodione, vinclozolin, and procymidone, are no longer approved. Authors must better specify the country/state and products available. They must be placed in an international context. Moreover, they also can include the FRAC CODE of the pathogen resistance and write something on the mode of action of fungicides.

‘Meanwhile, it also leads to the resistance of pathogens, which limits the wide application of fungicides [7].’ I suggest tying this sentence more closely to the previous one. Authors should also include some more references; one is not enough. Please, add international authors.

‘..a large number of substances considered safe have been successfully used to control postharvest gray mold damage in fruit and vegetables, such as CO [8], propionate [9], sodium silicate..’.

Authors should put some more references for each substance, one is not enough. Moreover, they could add just something about the mechanism of action against pathogens. Please, add international authors.

‘Some phenolic acids [11, 12] and essential oils [13] derived from plants are also applied to prevent gray mold damage’, please,  argue a little more. On which plant/fruit for example?

Explain better: ‘Amino acids are natural products of plants, being involved in many metabolic pathways within cells’

‘Besides that, they have the ability to induce disease resistance in plants.’ Add some references (Please, add international authors) and tie this sentence more closely to the previous one.

‘resistance of mango to anthrax’ are the authors sure of this??

‘might improve the resistance’, improve or not? How?

‘and other substances.’ Which ones? At least put to the family to which these compounds belong

‘Their metabolites are ethylene, polyamines (PA), etc.’ too much short

‘spermidine and spermidine have anti-aging effects’ too much ‘spermidine’

It should be underlined better that L-methionine or L-arginine are aminoacids. They should be introduced better when authors start to speak about aminoacids.

Rephrase in correct English ‘In general, these amino acids are mostly focused on the research of improving postharvest resistance of fruit and vegetable products, but their antifungal effects in vitro and inhibitory mechanisms have not been reported.’

Change ‘treatments on B. cinerea filament growth’ with ‘treatments on B. cinerea colony growth’

Please, avoid ‘and so on.’. Explain what work consists of.

I suggest changing ‘The research results might lay the foundation for understanding the antifungal capacity of L-methionine or L-arginine, and provide a novel perspective for the prevention and management of gray mold in post-harvest fruit and vegetables. ‘ with ‘This research could lay the ground for understanding the antifungal activity of L-methionine or L-arginine and provide potentially sustainable alternatives for the prevention and management of gray mold disease of fruit and vegetables in post-harvest.’

Please, check the paragraph titles, for example: ‘2.1. pathogen and spore suspension’. Check also other paragraphs.

In materials and methods please, adjust punctuation, grammar and sentence syntax. Many details must be added. Here just some examples

For example:

‘Test strain: Botrytis cinerea (No: BNCC185786) was purchased’, change with ‘The strain of Botrytis cinerea (No: BNCC185786) was purchased..’

‘was cultured at 28 for two weeks with the potato dextrose agar’ change with ‘was cultured at 28 for two weeks on potato dextrose agar medium (%x PDA)’

Change ‘The Spore suspension’ with ‘The spore suspension’. Please, check for similar errors.

It is not clear the part about spore suspension.

Change ‘(Met and Arg, ≥99%)’ with ‘(both  ≥99%)’

Change ‘Cherry tomatoes (Solanum lycopersicum var.saopola) ‘ with ‘Cherry tomatoes tomatoes var. Saopola‘

‘harvested at maturity’ Please, add some more details. How many days after for example flowering? (for example: fruits were harvested at maturity, 20 days after flowering).

2.3. Measurement of L-methionine and L-arginine on the mycelial growth of B. cinerea’ the title is not suitable. I suggest ‘2.3. Effect of L-methionine and L-arginine on the mycelial growth of B. cinerea’

‘described by Yang, Wang and Jiang [1].’. Please, follow the journal rules for citations in the text. Check for similar errors in the whole text.

‘were added to potato dextrose agar (90 mm diameter) to’ not exact. 90 mm is the diameter of a Petri dish. And PDA is poured into the dish. Please, correct the sentence and correct similar wrong sentences.

‘After the PDA solidified, a sterile punch was used to create a Botrytis cinerea cake from a pathogen culture. This cake’ Please, don’t use ‘cake’ and rephrase suitably and scientifically.

‘The cultures were sealed and incubated at a constant’. Petri dishes were sealed, not the cultures!

‘was assessed according to the previous method [28].’

‘of potato dextrose broth (PDB) medium containing a B. cinerea suspension (1 × 106 spores mL-1), separately’. It is not clear, ‘containing-separating’… explain better.

‘50 𝜇L of gray mold suspension’ change with ‘50 𝜇L of spore suspension’. However, the procedure is not clear.

‘aforementioned medium’ is the PDB? Explain better.

At the beginning of a sentence, never start with a number. Please, check this in the whole text.

Why do the authors write ‘previous method’ and then insert a reference never mentioned before? It is not clear. Please, explain better.

’ 100 𝜇L of spore suspension the fungal spore suspensions (1 ×106 spores mL-1) were added to a PDB’ too much ‘spore suspension’.

‘To eliminate potential back-ground absorbance effects from L-methionine and L-arginine, the absorbance values of L-methionine and L-arginine without B. cinerea were subtracted from the final results’. the author can refer to this as ‘blank’, making the sentence more readable. Change ‘were’ with ‘was’.

‘The MDA content of mycelium was determined after ‘ where?

Change ‘transferred to plastic containers ‘ with ‘transferred into plastic containers ‘

Change ‘was measured at 4 d after inoculation’ with ‘was measured 4 d after inoculation’

Statistical analysis: ‘SPSS Statistical software.’ Please, indicate software version.

Which ANOVA? One way, two ways? The p of ANOVA was significant? For example, for the effect of L-methionine and L-arginine treatments on mycelial growth, were the dishes arranged in a randomised design?

‘Analysis of variance (ANOVA) was carried out to determine the effects of the treatments, and those means were compared by Duncan's multiple range tests.’ This is true only in case of significant effect detected by ANOVA analysis. So, again, was the p of ANOVA significant (<0.05) for each parameter?

I suggest to change ‘All the experiments were performed in triplicate for each treatment and each replicate included three determinations.’ With ‘All the experiments were performed in triplicate (n=3).’ And delete ‘for each treatment and each replicate included three determinations.’

‘for each treatment and each replicate included three determinations.’ This detail should be explained very well in the ‘materials and methods’ section.

The differences between the means from three replicates with a value of p0.05 were considered statistically’ no sense. However, you have mentioned Duncan test, so it is a repetition. Authors must rephrase in one or two very clear sentences.

Figure 3, for example, lacks of statistical analysis. How can the authors discuss the results? Not mentioned what bars are. I suppose Standard deviation. However, it is not sufficient for discussion. What is ‘met’??

Legend is too imprecise. each data is an average of three experiments?

Legends Fig1 is not correct. C and B show also the effect of the treatment on colony growth, not only colony morphology. Statistical analysis is for supporting the data showed in the graphs!!

Not fully clear legend of Figure 4. ‘The leakage of proteins of B. cinerea was treated with different concentrations of L-methionine (C)and L-arginine (D).’

Also for nucleic acids there are different concentrations. Why is not reported?

Please, make one legend self-explenatory.

Fro Fig.4 I suggest to change ‘Different lowercase letters indicate significant differences among treatments at the same time (p0.05). Vertical bars represent standard deviations of the mean (n=3). ‘ with ‘Data are means of three experiments and vertical bars represent standard deviations. Different lowercase letters indicate significant differences among treatments within the same time, according to Duncan test (p0.05).’

Take this sentence as an example for all the others.

Legend of Figure 2 is too general and not self explenatory. Why ‘relative’?? what means? Not explained what ‘3h,6h…’ are, and also what ‘met’ and ‘arg’ are.

Avoid expressions as ‘was better’. It doesn’t mean anything.

For example ‘At the same concentration, the inhibitory effect of L-arginine on spore germination was better than that of L-methionine’. I would suggest ‘At the same concentration, the inhibitory effect of L-arginine on spore germination higher than that of L-methionine’.

errors in grammar, punctuation and syntax ‘Figure 8. Effects of L-methionine and L-arginine treatments on the lesion expansion of B. cinerea in postharvest cherry tomato. The disease spots of cherry tomato treated with L-methionine or L-argi-nine (A); The lesion diameter and disease incidence of cherry tomato fruit treated with L-methionine (B1) and L-arginine (B2).’

Comments on the Quality of English Language

Very difficult to understand.

There are many errors in grammar, punctuation and syntax.

Author Response

Response to Reviewer

Dear reviewer,

We very much appreciate the careful reading of our manuscript and valuable suggestions of the reviewer. We have carefully considered the comments and revised the manuscript accordingly. The comments can be summarized as follows:

  • Materials and methods. there are many unclear points: replicates/repetitions of the experiments.

We have provided a statement about the experimental replicates in the Materials and Methods section of the manuscript.

  • In my opinion, the statistical analysis section is too confusing and not clear and some figures lack of statistical analysis.  For this, I cannot review the results paragraph.

Our previous expression of the statistical analysis section was not accurate, and we have revised it in manuscript

  • The captions/legends of tables and figures must be rewritten more clearly, and they must be self-explanatory and written in comprehensible and correct English.

We have revised the wording of the legend and caption to make them more standardized and comprehensible.

Below, the authors can find in detail some suggestions for improving the manuscript and other critical issues I have found.

Abstract

  • ‘This study aimed to investigate the inhibitory effect of L-methionine or L-arginine on Botrytis cinerea.’ With’ This study aimed to investigate the effect of L-methionine or L-arginine on Botrytis cinereain vitro and on cherry tomato fruit’.’

We have made modifications according to the reviewer's suggestions.

  • ‘had obvious inhibitory effects’, please, avoid expressions like ‘obvious’. Not suitable for a scientific paper.

We replaced "obvious" with "significant" and simultaneously checked for similar errors.

Introduction

Page1:

  • Change ‘fungus on a wide range of fruit and’ with ‘fungus on a wide range of fruits and’

We have made modifications according to the reviewer's suggestions.

  • Change ‘plants with a high water content’ with ‘plants with a high-water content’

We have made modifications according to the reviewer's suggestions.

  • Do the authors refer to ‘plant’ or perishables fruit/vegetables with high-water content’?

There was indeed an issue with our expression. The correct wording should be ' fruits and vegetables with a high-water content’.

  • ‘It has the characteristics of strong adaptability, fast growth and spore production

We have made modifications according to the reviewer's suggestions

  • Adaptability to what? I suggest ‘Easy adaptability’

We have made modifications according to the reviewer's suggestions.

  • I suggest to change ‘Gray mold, caused by the pathogenic fungus  cinerea, leads to serious economic losses to global fruit and vegetable industry during harvesting, storage, processing, and transportation.’  with ‘Gray mold can lead to serious economic losses to global fruit and vegetable industry during harvesting, storage, processing, and transportation.’

We have made modifications according to the reviewer's suggestions.

  • Not truly correct ‘At present,  cinerea ranks among the top 10 most significant fungal plant diseases [3].’ B. cinerea is the microorganism, gray mold is the disease. Please, rephrase.

There was indeed an issue with our expression. The correct wording should be ' gray mold ranks among the top 10… ‘

  • ‘On one hand, it has ubiquitous harmfulness to fruit and vegetables;’ a concept already reported above.

‘…on the other hand, it is a model for in-depth analysis of the molecular mechanism of necrotrophic fungi pathogenicity and the development of environmentally friendly disease control strategies.

Your suggestion was indeed valuable, so we have made corrections in the manuscript to address the repetition.

  • Avoid such narration or rephrase. ‘Thus, many researchers pay attention to the study of  cinerea’ Please, check also throughout the text.

We have made modifications in the manuscript as per your advice and thoroughly checked the entire document.

  • ‘Owing to cost-effectiveness and ease of application, conventional synthetic chemical fungicides such as dicarboximides [5] and anilinopyrimidines [6] are commonly utilized to mitigate post-harvest decay in fruit and vegetables.’ Where? For example, in Europe, dicarboximide fungicides such as iprodione, vinclozolin, and procymidone, are no longer approved. Authors must better specify the country/state and products available. They must be placed in an international context. Moreover, they also can include the FRAC CODE of the pathogen resistance and write something on the mode of action of fungicides.

We have rephrased this sentence, adding specific details such as regions and available products, the mode of action of the fungicide.

  • ‘Meanwhile, it also leads to the resistance of pathogens, which limits the wide application of fungicides [7].’ I suggest tying this sentence more closely to the previous one. Authors should also include some more references; one is not enough. Please, add international authors.

We have rephrased this sentence, adding several references from some international authors.

Page2:

  • ‘..a large number of substances considered safe have been successfully used to control postharvest gray mold damage in fruit and vegetables, such as CO [8], propionate [9], sodium silicate..’.

Authors should put some more references for each substance, one is not enough. Moreover, they could add just something about the mechanism of action against pathogens. Please, add international authors.

We have rephrased this sentence, adding several references from some international authors.

18)     ‘Some phenolic acids [11, 12] and essential oils [13] derived from plants are also applied to prevent gray mold damage’, please,  argue a little more. On which plant/fruit for example?We have added explanations about the types and sources of phenolic acids and essential oils in the manuscript, along with the inclusion of international authors.19)     Explain better: ‘Amino acids are natural products of plants, being involved in many metabolic pathways within cells’We have incorporated explanations of amino acid functions in the manuscript.20)     ‘Besides that, they have the ability to induce disease resistance in plants.’ Add some references (Please, add international authors) and tie this sentence more closely to the previous one.We have created a tighter connection between the two sentences. Additionally, we have added references from international authors21)     ‘resistance of mango to anthrax’ are the authors sure of this??There was indeed an issue with our expression. The correct wording should be 'mango resistance to anthracnose.

  • ‘might improve the resistance’, improve or not? How?

'could improve the resistance ', this has been confirmed by the researchers, because Phenylalanine treatment induced the fruit defense response and the biosynthesis of antioxidant and antifungal flavonoids, which can effectively control fungal growth and disease development. Treatment with phenylalanine also enhanced chilling tolerance of mango fruit through regulation of metabolic and defense-related pathways, maintaining high levels of flavonoids, and antioxidants enzyme activity, and reducing H2O2 content, lipid peroxidation, and volatile aldehydes. I have made the necessary correction in the manuscript.

  • ‘and other substances.’ Which ones? At least put to the family to which these compounds belong

We have added detailed information about "other substances" in the manuscript.

  • ‘Their metabolites are ethylene, polyamines (PA), etc.’ too much short

We provided a more detailed explanation of the types of metabolites in the manuscript, avoiding the use of "etc." We have also thoroughly reviewed the entire document.

  • ‘spermidine and spermidine have anti-aging effects’ too much ‘spermidine’

There was indeed an issue with our expression. The correct wording should be ' spermidine and spermine ‘.

  • It should be underlined better that L-methionine or L-arginine are aminoacids. They should be introduced better when authors start to speak about aminoacids.

We have made corresponding modifications in the manuscript.

  • Rephrase in correct English ‘In general, these amino acids are mostly focused on the research of improving postharvest resistance of fruit and vegetable products, but their antifungal effects in vitro and inhibitory mechanisms have not been reported.’

We have revised the sentence to ‘In general, the research of these two amino acids (L-methionine or L-arginine) mainly focused on improving the postharvest resistance of fruits and vegetables, but their antifungal activities and inhibitory mechanisms in vitro have not been reported'.

  • Change ‘treatments on  cinerea filament growth’ with ‘treatments on B. cinerea colony growth’

We have made corresponding modifications in the manuscript.

  • Please, avoid ‘and so on.’. Explain what work consists of.

We have added more explanations in the manuscript to avoid using "so on" for clarification, and we have checked other parts of the entire document.

  • I suggest changing ‘The research results might lay the foundation for understanding the antifungal capacity of L-methionine or L-arginine, and provide a novel perspective for the prevention and management of gray mold in post-harvest fruit and vegetables. ‘ with ‘This research could lay the ground for understanding the antifungal activity of L-methionine or L-arginine and provide potentially sustainable alternatives for the prevention and management of gray mold disease of fruit and vegetables in post-harvest.’

We have made modifications to the manuscript based on your suggestions.

Page2: Section 2.1

  • Please, check the paragraph titles, for example: ‘1. pathogen and spore suspension’.Check also other paragraphs.

We replaced " 2.1. pathogen and spore suspension " 2.1. Source of Fungi and Preparation of Spore Suspension"

  • In materials and methods please, adjust punctuation, grammar and sentence syntax. Many details must be added. Here just some examples

We have made modifications in the manuscript as per your advice and thoroughly checked the entire document.

For example:

Section 2.1

  • ‘Test strain: Botrytis cinerea (No: BNCC185786) was purchased’, change with ‘The strain of Botrytis cinerea (No: BNCC185786) was purchased.’

We have made modifications to the manuscript based on your suggestions.

  • ‘was cultured at 28 ℃ for two weeks with the potato dextrose agar’ change with ‘was cultured at 28 ℃ for two weeks on potato dextrose agar medium (%x PDA)’

The errors have been corrected as per the suggestions.

Potato dextrose agar (PDA) medium was prepared according to classic formula.

  • Change ‘The Spore suspension’ with ‘The spore suspension’. Please, check for similar errors.

We replaced " The Spore suspension " with " The spore suspension " and simultaneously checked for similar errors.

  • It is not clear the part about spore suspension.

We have modified the part about preparation of spore suspension

Page3: Section 2.2

  • Change ‘(Met and Arg, ≥99%)’ with ‘(both  ≥99%)’

We have made modifications to the manuscript based on your suggestions

  • Change ‘Cherry tomatoes (Solanum lycopersicum var.saopola) ‘ with ‘Cherry tomatoes tomatoes var. Saopola‘

We have made modifications to the manuscript based on your suggestions

  • ‘harvested at maturity’ Please, add some more details. How many days after for example flowering? (for example: fruits were harvested at maturity, 20 days after flowering).

Thank you very much for your careful review of our work. Cherry tomatoes were harvested at commercial maturity. They are organic, and they were cleaned with sodium hypochlorite before inoculation.

Page3: Section 2.3

  • 3. Measurement of L-methionine and L-arginine on the mycelial growth of B. cinerea’ the titleis not suitable. I suggest ‘2.3. Effect of L-methionine and L-arginine on the mycelial growth of B. cinerea’

We have made modifications to the manuscript based on your suggestions.

  • ‘described by Yang, Wang and Jiang [1].’. Please, follow the journal rules for citations in the text. Check for similar errors in the whole text.

There was indeed an issue with our expression. We have corrected the manuscript according to the citation rules of the journal.

  • ‘were added to potato dextrose agar (90 mm diameter) to’ not exact. 90 mm is the diameter of a Petri dish. And PDA is poured into the dish. Please, correct the sentence and correct similar wrong sentences.

We have made modifications to the manuscript based on your suggestions.

  • ‘After the PDA solidified, a sterile punch was used to create a Botrytis cinerea cake from a pathogen culture. This cake’ Please, don’t use ‘cake’ and rephrase suitably and scientifically.

We replaced " Botrytis cinerea cake " with " the mycelial disks " and simultaneously checked for similar errors.

  • ‘The cultures were sealed and incubated at a constant’. Petri dishes were sealed, not the cultures!

We have made modifications to the manuscript based on your suggestions.

  • ‘was assessed according to the previous method [28].’

We replaced " previous method " with " previously reported methods "

  • ‘of potato dextrose broth (PDB) medium containing a  cinerea suspension (1 × 106 spores mL-1), separately’. It is not clear, ‘containing-separating’… explain better.

We have made the corresponding changes in the manuscript.

‘L-methionine solutions at concentrations of 0, 0.5, 5, 50, and 100 mmol L-1, and L-arginine solutions at concentrations of 0, 6.25, 12.5, 25, and 50 mmol L-1 were prepared using sterile 20 mL of potato dextrose broth (PDB) medium, containing-separating a B. cinerea suspension (1 × 106 spores mL-1).’

  • ‘50 ?L of gray mold suspension’ change with ‘50 ?L of spore suspension’. However, the procedure is not clear.

We have made modifications to the manuscript based on your suggestions.

We have revised the section on spore suspension preparation and provided explanations in title2.1.

  • ‘aforementioned medium’ is the PDB? Explain better.

We replaced " aforementioned medium " with " PDB " and simultaneously checked for similar errors.

Page3: Section 2.4.1.

  • At the beginning of a sentence, never start with a number. Please, check this in the whole text.

We have made modifications to the manuscript based on your suggestions.

  • Why do the authors write ‘previous method’ and then insert a reference never mentioned before? It is not clear. Please, explain better.

There was indeed an issue with our expression. We replaced " previous method " with " previously reported methods "

Page3: Section 2.4.2

  • ’ 100 ?L of spore suspension the fungal spore suspensions (1 ×106 spores mL-1) were added to a PDB’ too much ‘spore suspension’.

It was an oversight in our experimental procedure. Maybe a little too much spore suspension. We will pay attention to this matter in future experiments.

Page4: Section 2.4.2

  • ‘To eliminate potential back-ground absorbance effects from L-methionine and L-arginine, the absorbance values of L-methionine and L-arginine without  cinerea were subtracted from the final results’. the author can refer to this as ‘blank’, making the sentence more readable. Change ‘were’ with ‘was’.

We have made modifications to the manuscript based on your suggestions.

Page4: Section 2.4.3

  • ‘The MDA content of mycelium was determined after ‘ where?

It was cultured on a rotary incubator, so we added this part in the manuscript.

Page5: Section 2.7

  • Change ‘transferred to plastic containers’ with ‘transferred into plastic containers

We have made modifications to the manuscript based on your suggestions.

  • Change ‘was measured at 4 d after inoculation’ with ‘was measured 4 d after inoculation’

We have made modifications to the manuscript based on your suggestions.

Page5: Section 2.8

  • Statistical analysis: ‘SPSS Statistical software.’ Please, indicate software version.

We added the version of SPSS in the manuscript.

  • Which ANOVA? One way, two ways? The p of ANOVA was significant? For example, for the effect of L-methionine and L-arginine treatments on mycelial growth, were the dishes arranged in a randomised design?

We have made corresponding modifications to the statistical analysis of the manuscript and added relevant details. We employed a one-way analysis of variance (ANOVA) for the experimental results, followed by Duncan's multiple range test, with a confidence level of 95% (p <0.05).

Our Petri dishes were arranged in a randomized design.

  • ‘Analysis of variance (ANOVA) was carried out to determine the effects of the treatments, and those means were compared by Duncan's multiple range tests.’ This is true only in case of significant effect detected by ANOVA analysis. So, again, was the p of ANOVA significant (<0.05) for each parameter?

In our experimental results, not all parameters showed a significant analysis result of variance (ANOVA) in the whole experiment, but as long as the result of ANOVA is p less than 0.05 (p<0.05), the difference is significant.

  • I suggest to change ‘All the experiments were performed in triplicate for each treatment and each replicate included three determinations.’ With ‘All the experiments were performed in triplicate (n=3).’ And delete ‘for each treatment and each replicate included three determinations.’

We have made modifications based on your suggestion. Your feedback is highly comprehensive, and as a result, we have augmented the materials and methods section accordingly.

  • ‘for each treatment and each replicate included three determinations.’ This detail should be explained very well in the ‘materials and methods’ section.

According to your suggestion, we have added more explanations in the ‘materials and methods’ section.

  • ‘The differences between the means from three replicates with a value of p<05 were considered statistically’ no sense. However, you have mentioned Duncan test, so it is a repetition. Authors must rephrase in one or two very clear sentences.

There was a logical issue in our previous expression, so we have rewritten this sentence.

‘To compare the different treatments, Duncan’s multiple range test was employed at a confidence level of 95% (p < 0.05)’

Page7: Section 3.2

  • Figure 3, for example, lacks of statistical analysis. How can the authors discuss the results? Not mentioned what bars are. I suppose Standard deviation. However, it is not sufficient for discussion. What is ‘met’??

We have supplemented the statistical analysis for Figure 3 and added explanations for the legend and captions. The bars represent standard deviation. ‘met’ is the abbreviation of L-arginine, and we have eliminated the abbreviations 'Met' and 'Arg', using the full terms and maintaining consistency throughout the text.

  • Legend is too imprecise. each data is an average of three experiments?

We have revised the legends and captions. The data are the means of three replications.

Page5: Section 3.1

  • Legends Fig1 is not correct. C and B show also the effect of the treatment on colony growth, not only colony morphology. Statistical analysis is for supporting the data showed in the graphs!!

We have made corrections in the manuscript as per your suggestions, ensuring proper labeling.

Page8: Figure 4

  • Not fully clear legend of Figure 4. ‘The leakage of proteins of  cinerea was treated with different concentrations of L-methionine (C)and L-arginine (D).’

Also for nucleic acids there are different concentrations. Why is not reported?

Please, make one legend self-explenatory.

We have supplemented and modified the figure caption as follows: ' Effects of L-methionine and L-arginine treatments on nucleic acids and proteins leakage of B. cinerea. The leakage of nucleic acids of B. cinerea treated with (A) L-methionine and (B) L-arginine. The leakage of proteins of B. cinerea treated with (C) L-methionine and (D) L-arginine. Data are means of three experiments and vertical bars represent standard deviations. Different lowercase letters in-dicate significant differences among treatments within the same time, according to Duncan test (p<0.05).'

  • For Fig.4 I suggest to change ‘Different lowercase letters indicate significant differences among treatments at the same time (p<05). Vertical bars represent standard deviations of the mean (n=3). ‘ with ‘Data are means of three experiments and vertical bars represent standard deviations. Different lowercase letters indicate significant differences among treatments within the same time, according to Duncan test (p<0.05).’

Take this sentence as an example for all the others.

We have made modifications in the manuscript as per your advice and thoroughly checked the entire document.

Page8: Figure 2

  • Legend of Figure 2 is too general and not self explenatory. Why ‘relative’?? what means? Not explained what ‘3h,6h…’ are, and also what ‘met’ and ‘arg’ are.

We have modified the legend for Figure 2, added explanations for time, and avoided the use of abbreviations.

‘Effects of L-methionine and L-arginine treatments on spore germination of B. cinerea. Photographs (A) of the germinating spores of B. cinerea cultured with L-methionine and L-arginine at 3h, 6h, 9h, 12h, and 15h. Statistical analysis of spore germination rate of B. cinerea treated with (B) L-methionine and (C) L-arginine.’

Page7: Section 3.1

  • Avoid expressions as ‘was better’. It doesn’t mean anything.

For example ‘At the same concentration, the inhibitory effect of L-arginine on spore germination was better than that of L-methionine’. I would suggest ‘At the same concentration, the inhibitory effect of L-arginine on spore germination higher than that of L-methionine’.

We have made modifications according to the reviewer's suggestions.

Page12: Figure 8

  • errors in grammar, punctuation and syntax ‘Figure 8. Effects of L-methionine and L-arginine treatments on the lesion expansion of  cinerea in postharvest cherry tomato. The disease spots of cherry tomato treated with L-methionine or L-argi-nine (A); The lesion diameter and disease incidence of cherry tomato fruit treated with L-methionine (B1) and L-arginine (B2).’

We have rewritten the content. ‘Effects of L-methionine and L-arginine treatments on the lesion expansion of B. cinerea in postharvest cherry tomatoes. The disease spots of cherry tomatoes treated with (A) L-methionine and (B) L-arginine. Decay diameter and disease incidence in cherry tomatoes treated with (C) L-methionine and (D) L-arginine. Data are means of three experiments and vertical bars represent standard deviations. Different lowercase letters indicate significant differences among treatments within the same time, according to Duncan test (p<0.05).

Reviewer 2 Report

Comments and Suggestions for Authors

„Cherry tomatoes (Solanum lycopersicum var.saopola) were harvested at maturity, and purchased from Huilong Farmers Market located in Yuci District, Jinzhong City, Shanxi Province, China.” - Were the tomatoes used grown without the use of fungicides and were they organic? The presence of fungicide residues could have influenced the test results.

„After the PDA solidified, a sterile punch was used to create a Botrytis cinerea cake from a pathogen culture. This cake was then put in the middle of each Petri dish” - what was the initial diameter of B. cinerea?

What is the practical application of the results obtained in the context of costs?

Author Response

Response to Reviewer

Dear reviewer,

We very much appreciate the careful reading of our manuscript and valuable suggestions of the reviewer. We have carefully considered the comments and have revised the manuscript accordingly. The comments can be summarized as follows:

Page3: Section 2.2

  • Cherry tomatoes (Solanum lycopersicum var.saopola) were harvested at maturity, and purchased from Huilong Farmers Market located in Yuci District, Jinzhong City, Shanxi Province, China.” - Were the tomatoes used grown without the use of fungicides and were they organic? The presence of fungicide residues could have influenced the test results.

Thank you very much for your careful review of our work. Cherry tomatoes were harvested at commercial maturity. They were cleaned with sodium hypochlorite before inoculation.

Page3: Section 2.3

  • „After the PDA solidified, a sterile punch was used to create a Botrytis cinerea cake from a pathogen culture. This cake was then put in the middle of each Petri dish” - what was the initial diameter of B. cinerea?

The initial diameter of B. cinerea is 13 mm. We have incorporated this information into the manuscript.

  • What is the practical application of the results obtained in the context of costs?

L-methionine and L-arginine are two very common amino acids, not expensive compared with other biologics (e.g., plant essential oil, antibiotic and enzyme). They are healthier and more environmentally friendly than chemical fungicides. So they have potential practical value in production.

Reviewer 3 Report

Comments and Suggestions for Authors

In the presented manuscript, authors showed the negative effect of amino acids methionine and arginine on phytopathogen Botrytis cinerea, while arginine had a markedly stronger effect than methionine. Importantly, while arginine had a greater negative effect on the fungus growth in an artificial medium, the protective effect of methionine on tomato fruit was not weaker than that of arginine

I believe that the presented article contains two main problems: (1) data on the negative effect of these amino acids, especially arginine, have already been presented several times in the literature and (2) very high concentrations of amino acids (up to 100 mM) were used in the experiments. Several articles in which the mentioned effect of the studied amino acids has been previously described are present in the reference list (Hasabi et al, 2014; Zhang et al., 2017, Zheng et al., 2011) but are not discussed by the authors  in relation to the previously described effects on B. cinerea. In this regard, the novelty of the presented study should be clearly justified. Also, the feasibility of using such high concentrations in experiments and practical applications should be clearly stated.

There are many technical flaws in the article. The following are some examples.

1) In the sentence “In general, these AMINO ACIDS ARE MOSTLY FOCUSED ON THE RESEARCH of improving postharvest resistance of fruit and vegetable productsbut their antifungal effects in vitro and inhibitory mechanisms have not been reported”. - The research is focused, not amino acids?

2) Please, clarify, what means “The Spore suspension was prepared for field use”

3) Measurement of L-methionine and L-arginine on the mycelial growth of B. cinerea - Measurement of the L-methionine and L-arginine EFFECT on the mycelial growth of B. cinerea?

4) The effects of L-methionine and L-arginine on mycelial growth of B. cinerea were shown in Figure 1. It should be “are shown”.

5) Figure 2. (A) Relative photographs of the germinating spores of B. cinerea.  It should be “Representative photographs…”

6) Fungal spores have remarkable tolerance and Adaptability to various – “A” in “adaptability” should not be capital.

7) It is written: “However, the samples treated with low-concentration L-methionine or L-arginine showed partial bright red fluorescence, and the samples treated with high-concentration of L-methionine or L-arginine displayed a large area of strong red fluorescence”. – Indeed, both concentration are high.

Comments on the Quality of English Language

The text contains meny technical errors.

Author Response

Response to Reviewer

Dear reviewer,

We very much appreciate the careful reading of our manuscript and valuable suggestions of the reviewer. We have carefully considered the comments and have revised the manuscript accordingly. The comments can be summarized as follows:

I believe that the presented article contains two main problems: (1) data on the negative effect of these amino acids, especially arginine, have already been presented several times in the literature and (2) very high concentrations of amino acids (up to 100 mM) were used in the experiments. Several articles in which the mentioned effect of the studied amino acids has been previously described are present in the reference list (Hasabi et al, 2014; Zhang et al., 2017, Zheng et al., 2011) but are not discussed by the authors in relation to the previously described effects on B. cinerea. In this regard, the novelty of the presented study should be clearly justified. Also, the feasibility of using such high concentrations in experiments and practical applications should be clearly stated.

Thank you for your comment on our work.

For the first problem, although the negative effect of these amino acids at high concentrations, especially arginine, have already been presented in some literatures. Our research idea is different, our study focuses on the direct antifungal effect of amino acids rather than their induction of disease resistance. Also, our treatment is different, we inoculate the pathogen first and then treat it with amino acids.

For the second problem, compare the results of previous studies (Hasabi et al, 2014; Zhang et al., 2017, Zheng et al., 2011), we discussed effects of the two amino acids on B. cinerea in discussion to justify the novelty of the presented study. 

And finally, L-methionine and L-arginine are two very common amino acids, not expensive compared with other biologics (e.g., plant essential oil, antibiotic and enzyme). They are healthier and more environmentally friendly than chemical fungicides. So, they have potential practical value in production.

Introduction

Page2:

1) In the sentence “In general, these AMINO ACIDS ARE MOSTLY FOCUSED ON THE RESEARCH of improving postharvest resistance of fruit and vegetable products,but their antifungal effects in vitro and inhibitory mechanisms have not been reported”. - The research is focused, not amino acids?

There was indeed an issue with our previous expression. We have revised the sentence to 'In general, the research of these two amino acids (L-methionine or L-arginine) mainly focused on improving the postharvest resistance of fruits and vegetables, but their antifungal activities and inhibitory mechanisms in vitro have not been reported'.

Page2: Section 2.1

2) Please, clarify, what means “The Spore suspension was prepared for field use”

It was a clerical error. It has been revised. What we mean is ‘used immediately after preparation’

Page3: Section 2.3

3) Measurement of L-methionine and L-arginine on the mycelial growth of B. cinerea - Measurement of the L-methionine and L-arginine EFFECT on the mycelial growth of B. cinerea?

We made corresponding changes to the manuscript.

Page5: Section 3.1

4)The effects of L-methionine and L-arginine on mycelial growth of B. cinerea were shown in Figure 1. It should be “are shown”.

We have revised the manuscript as per your request.

Page7: Figure 2

5) Figure 2. (A) Relative photographs of the germinating spores of B. cinerea.  It should be “Representative photographs…”

We have made the correction.

Page7:

6) Fungal spores have remarkable tolerance and Adaptability to various – “A” in “adaptability” should not be capital.

‘Adaptability ‘was replaced with ‘adaptability’

Page10: Section 3.3

7) It is written: “However, the samples treated with low-concentration L-methionine or L-arginine showed partial bright red fluorescence, and the samples treated with high-concentration of L-methionine or L-arginine displayed a large area of strong red fluorescence”. – Indeed, both concentration are high‘.

Although their fluorescence treated with both concentrations are high, however, there is still a significant difference in brightness. To show our results more clearly, we have made the modification in the manuscript, and the new sentence is: However, the samples treated with 50 mmol L-1 L-methionine or 25 mmol L-1 L-arginine showed small parts of bright red fluorescence, while those treated with 100 mmol L-1 L-methionine or 50 mmol L-1 L-arginine displayed a large area of intense red fluorescence.

Round 2

Reviewer 3 Report

Comments and Suggestions for Authors

All suggestions are accepted by authors, all mistakes are corrected.